# Mortality on the UNOS Waitlist for Patients with Autoimmune Liver Disease

**DOI:** 10.3390/jcm9020319

**Published:** 2020-01-23

**Authors:** Jaspreet S. Suri, Christopher J. Danford, Vilas Patwardhan, Alan Bonder

**Affiliations:** Liver Center, Division of Gastroenterology and Hepatology, Beth Israel Deaconess Medical Center, Harvard Medical School, Boston, MA 02215, USA

**Keywords:** autoimmune liver disease, autoimmune hepatitis, primary sclerosing cholangitis, primary biliary cholangitis, liver transplant, mortality, waitlist

## Abstract

Background: Outcomes on the liver transplant waitlist can vary by etiology. Our aim is to investigate differences in waitlist mortality of autoimmune hepatitis (AIH), primary biliary cholangitis (PBC), and primary sclerosing cholangitis (PSC) using the United Network for Organ Sharing (UNOS) database. Methods: We identified patients who were listed for liver transplantation from 1987 to 2016 with a primary diagnosis of AIH, PBC, or PSC. We excluded patients with overlap syndromes, acute hepatic necrosis, missing data, and those who were children. The primary outcome was death or removal from the waitlist due to clinical deterioration. We compared waitlist survival using competing risk analysis. Results: Between 1987 and 2016, there were 7412 patients listed for liver transplant due to AIH, 8119 for PBC, and 10,901 for PSC. Patients with AIH were younger, more likely to be diabetic, and had higher listing model for end-stage liver disease (MELD) scores compared to PBC and PSC patients. Patients with PBC and AIH were more likely to be removed from the waitlist due to death or clinical deterioration. On competing risk analysis, AIH patients had a similar risk of being removed from the waitlist compared to those with PBC (subdistribution hazard ratio (SHR) 0.94, 95% CI 0.85–1.03) and higher risk of removal compared to those with PSC (SHR 0.8, 95% CI 0.72 to 0.89). Conclusion: Autoimmune hepatitis carries a similar risk of waitlist removal to PBC and a higher risk than PSC. The etiology of this disparity is not entirely clear and deserves further investigation.

## 1. Introduction

Autoimmune liver diseases, including autoimmune hepatitis (AIH), primary biliary cholangitis (PBC), and primary sclerosing cholangitis (PSC), account for up to 24% of liver transplants in the United States and Europe [1]. Four to six percent of these are secondary to AIH, a rare disease with a heterogenous presentation [2]. Post-transplant outcomes are significantly worse for AIH compared to PBC and PSC, with higher risk of death or need for re-transplantation. Therefore, despite all being autoimmune in nature and overlaps existing between the three diseases, AIH, PBC, and PSC each have distinct clinical courses, treatments, and outcomes.

Although 1767 candidates in the United States died while on the waitlist in 2013 with an additional 1223 being removed for being too ill [3], evidence exists that these outcomes on the liver transplant waitlist vary by autoimmune etiology. For example, patients listed for PBC have higher waitlist mortality than those with PSC, as shown by one study reviewing the United Network for Organ Sharing (UNOS) database between 2002 and 2013 [4]. Another group who also reviewed the UNOS database for the years 2002 to 2016 demonstrated that waitlist mortality is lower in patients listed for AIH than those listed for nonalcoholic steatohepatitis (NASH). Although they included comparisons to alcoholic liver disease and cryptogenic cirrhosis, comparisons to other autoimmune liver diseases are lacking [5].

Here we focus on AIH and its waitlist mortality in comparison to PBC and PSC via review of the United Network for Organ Sharing (UNOS) database. We hypothesize that, given the widespread availability of corticosteroids and their effectiveness in treatment of AIH, this disease will have lower waitlist mortality than PSC. We also sought to confirm the relatively high waitlist mortality of PBC and investigate how it compares to AIH.

## 2. Materials and Methods

### 2.1. Study Population

We used the UNOS Organ Procurement and Transplant Network (OPTN) database to identify all patients who were listed for their first liver transplant from 1987 to 2016. We included only those patients with a primary listing diagnosis of AIH, PBC, or PSC and excluded patients with overlap syndromes, missing data, children (age < 18 years), those listed for combined liver and kidney transplant, and those with acute hepatic necrosis (as this diagnosis is not etiology specific in the UNOS database).

### 2.2. Outcomes

The primary outcome was waitlist survival using the composite outcome of death or removal for clinical deterioration (UNOS removal codes 5, 8, and 13). We compared waitlist survival among the three diseases using competing risk analysis with liver transplantation as a competing risk.

### 2.3. Statistical Analysis

Demographics and clinical characteristics of patients listed for liver transplant for AIH, PBC, and PSC were compared using one-way ANOVA for continuous variables and the chi-squared test for categorical variables. Kaplan–Meier curves and Cox regression models were used to compare patient survival on the waitlist for AIH, PBC, and PSC patients. We adjusted for recipient characteristics including age, sex, diabetes diagnosis, albumin at listing, blood type, model for end-stage liver disease (MELD) score at listing, and region in which the patient was listed. Competing risk analysis was used to evaluate the cumulative incidence of death or delisting for deterioration with liver transplant as a competing risk. Trends over time were analyzed using the Cochran–Armitage test. All analyses were performed using Stata 11.2 (College Station, TX, USA).

## 3. Results

Between 1987 and 2016, there were 26,432 patients listed for liver transplant in the United States for autoimmune liver disease, with 7412 for AIH, 8119 for PBC, and 10,901 for PSC (Figure 1). Baseline demographics for the respective groups are displayed in Table 1. Patients with AIH were younger (44.6 ± 16.3 years) compared to PBC (54.9 ± 10.0) and PSC (45.6 ± 14.3) patients (*p* < 0.001), and more likely to be diabetic (20.4%) compared to PBC (15.9%) and PSC (12.1%) patients (*p* < 0.001). Patients with AIH also had a higher listing MELD (18.9 ± 9.5) compared to PBC (17.1 ± 7.9) and PSC (16.4 ± 7.9) (*p* < 0.001).

### 3.1. Waitlist Survival

Patients with PBC and AIH were more likely to be removed from the waitlist for death or clinical deterioration (17.5% and 15%, respectively) than PSC (10.2%) patients (Table 2). The cause of death was unknown in the majority of patients, irrespective of the listing disease. Patients with AIH were less likely to ultimately be transplanted (52.3%) than PBC (59.1%) and PSC (65.9%) (*p* < 0.001) (Table 1). Median waitlist time was 7.5 months (interquartile range (IQR) 1.2–27.5) for AIH, 8.5 months (IQR 2.3–23.7) for PBC, and 7.7 months (IQR 2.3–22.9) for PSC.

### 3.2. Cox Regression and Competing Risk Analysis

On multivariate cox regression analysis, PBC was more likely to be removed for death or clinical deterioration in comparison to AIH (Table 3). While on competing risk analysis with transplant as a competing risk, patients with AIH had a similar risk of being removed from the waitlist for death or clinical deterioration compared to PBC (subdistribution hazard ratio (SHR) 0.94, 95% confidence interval (CI) 0.85–1.03), and higher risk of removal compared to PSC (SHR 0.8, 95% CI 0.72–0.89) (Table 4, Figure 2).

### 3.3. Temporal Trends

We also examined temporal trends in liver transplant listing by autoimmune etiology. AIH as a proportion of autoimmune liver disease listings has generally increased over time from 26.5% in the period from 1987 to 1995, to 30% in 2008 to 2016. On the other hand, listing for PBC has generally decreased from 37.6% in 1987 to 1995, to 27.3% in 2008 to 2016. Listing for PSC increased over this time period, as shown in Table 5. However, these trends were not significant (*p* = 0.175).

## 4. Discussion

Listing for an orthotopic liver transplant secondary to autoimmune liver diseases has decreased over time. Once listed, outcomes can vary and authors have previously compared those of PBC and PSC. Despite this, studies comparing waitlist outcomes of AIH to the other autoimmune liver diseases have been limited. In this study, we analyzed the UNOS database to uncover trends on removal for death or clinical deterioration from the liver transplant waitlist in patients listed for autoimmune liver disease. Our novel finding, presented via the competing risk analysis, shows that AIH carries a similar risk of waitlist removal due to death or deterioration compared to PBC and higher risk compared to PSC after accounting for factors such as age, sex, listing MELD, listing albumin, UNOS region, and blood type.

The reason for higher adverse waitlist outcomes in AIH and PBC compared to PSC is not entirely clear. Unfortunately, cause of death is not accurately reported in the UNOS database, with over 60% reported as “other, unknown, or multiorgan failure” amongst all the autoimmune etiologies (Table 2). Of the causes delineated, infection was the most commonly reported amongst all three autoimmune conditions, with cardiac causes and liver-related events rounding out the top three most documented reasons for death. We hypothesize that older age among PBC patients and higher rates of diabetes among AIH patients compared to PSC may contribute to the higher rate of cardiac complications reported. Additionally, PBC and AIH patients were listed with a higher MELD score compared to PSC patients, indicating that more advanced liver disease may also play a role in the higher rate of adverse waitlist outcomes. On multivariate competing risk analysis, although female sex, age, diagnosis of diabetes, and listing MELD did contribute to increased risk of waitlist death or delisting for deterioration, AIH patients remained at higher risk compared to PSC patients by nature of diagnosis alone (Table 4).

Furthermore, we postulate that poor outcomes for patients with AIH on the waitlist may be related to complications of management or the disease itself. Immunosuppression with azathioprine or steroids puts patients at significant risk for serious infections. Not only that, one population-based study showed significantly higher rates of hepatobiliary, non-melanoma skin, and hematologic cancers in the AIH population [6]. Poorly controlled disease in AIH could also remove patients from the list secondary to a significant flare as 10% to 15% of patients are refractory to standard treatment from non-compliance, partial compliance, or a true non-response [7]. This is further complicated by comorbid conditions that preclude the use of therapeutic doses of immunosuppression. As a result, we are left with the AIH patient group (with higher listing MELD scores and increased rates of complications while on the list) not being transplanted as much as the PBC or PSC groups (Table 1).

Our findings also confirm that PBC (17.5%) has a higher waitlist mortality compared to PSC (10.2%), in addition to AIH with 15% [4,8]. In a study including 100 PBC patients and 76 PSC patients listed for transplant in Austria, waitlist mortality was significantly higher in PBC patients compared to PSC patients [9]. In a different study using the UNOS database, Singal et al. also reported significantly higher mortality among PBC patients (21.6%) compared to PSC patients (12.7%) independent of the listing MELD score [4].

PSC has been shown to have lower rates of death and removal from the waitlist for deterioration compared to other forms of chronic liver disease in general (13.6% vs. 20.5%, *p* < 0.001) with an adjusted hazard ratio of 0.72 (CI 0.66–0.79, *p* < 0.001). A plausible explanation was thought to be the lower risk of developing complications from portal hypertension in patients listed with PSC [10]. Surprisingly, this trend was present even with adjustments for MELD score, complications of portal hypertension at the time of listing, differential rates of liver donor transplant, and allotment of MELD exception points. Singal et al. showed that adjusted waitlist mortality did not differ significantly between PBC and PSC if patients with PSC receiving exception points were removed [4]. Another study showed that waitlist candidates with PSC and a history of bacterial cholangitis did not have increased risk of mortality over PSC patients without cholangitis, which is a major indication for allowing exception points [11]. Therefore, it is plausible that the difference in mortality seen on the waitlist between PBC and PSC is strongly influenced by the allotment of MELD exception points—the process of which may need to be reevaluated based on this evidence.

As our temporal data shows, PSC has overtaken AIH and PBC by proportion of waitlisted patients for autoimmune liver disease. The decline of PBC as a reason for listing is likely influenced by the rise of ursodeoxycholic acid as a validated and effective treatment since 1994 [12]. Interestingly, the effect is not the same for PSC, where the use of ursodeoxycholic acid has also garnered interest, although without the same level of confidence in the benefits [13]. This decline is also probably influencing the decrease in listing for autoimmune liver disease overall, as seen between 1996–2007 and 2008–2016. Despite the effectiveness of corticosteroids and immunomodulators, AIH as a listing diagnosis has slightly increased in proportion instead of decreasing. This again highlights the difficulty of successfully treating and sustaining remission in AIH. Broadly speaking, these findings must be interpreted carefully as we do not include overlap syndromes or other autoimmune liver diseases for analysis. Additionally, there are important gaps in the data that hinder our completeness, such as the lack of information regarding incidence and prevalence rates for AIH. Overall, even though the differences here were statistically not significant, our findings are similar to those of Webb et al., who went further to include the changing trends in ethnicity and gender as well for a similar cohort [14].

The strengths of our study include its analysis of a large-scale database of transplant candidates, over an expansive time period, allowing for revelation of trends that may be generalizable to other transplant networks and the autoimmune liver disease population in general. Secondly, we used competing risk analysis, whereas other survival models may provide overestimation of the risk of death or deterioration and underestimation of liver transplant probability. Additionally, no prior study to our knowledge has looked specifically at the individual autoimmune liver diseases in comparison to each other on this scale, specifically commenting on AIH. We also used this opportunity to confirm trends noted by other groups who also retrospectively analyzed this UNOS database.

Limitations of this study are shared by some of the studies that have come before looking at the UNOS database; namely its retrospective nature and the lack of granularity inherent in a large, nationwide database. This limitation in the data that is available for review creates difficulties in drawing more conclusive arguments. For example, clarifying the cause of death or clinical deterioration could prove useful for future work. Data on waitlist mortality across all liver diseases remain scant and deserves more attention.

## 5. Conclusions

In conclusion, continued study is necessary to clearly determine what is driving these differences in waitlist outcomes amongst the autoimmune liver conditions. More information on liver and non-liver-related decompensations in the UNOS database may be of help. Additionally, it may be useful to review our process of awarding MELD exception points and their utility, and debate on who should qualify. By addressing these two points, the future of allocation for liver transplantation could undergo significant change.

## Figures and Tables

**Figure 1 jcm-09-00319-f001:**
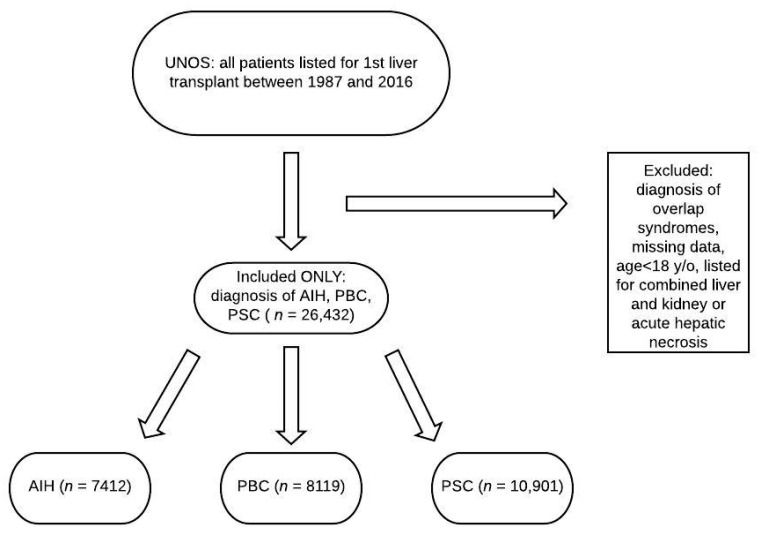
Flowchart of patients reviewed. UNOS: United Network for Organ Sharing; AIH: autoimmune hepatitis; PBC: primary biliary cholangitis; PSC: primary sclerosing cholangitis.

**Figure 2 jcm-09-00319-f002:**
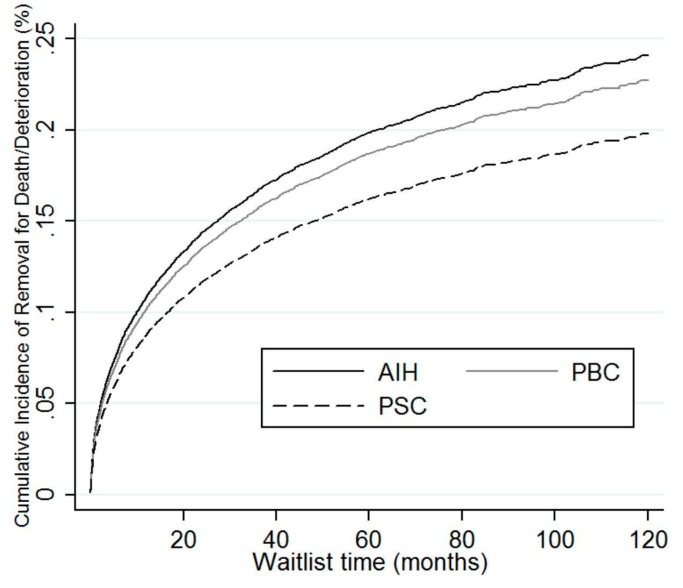
Competing risk regression of AIH, PBC, and PSC demonstrating relative removal from waitlist for death or deterioration. AIH: autoimmune hepatitis; PBC: primary biliary cholangitis; PSC: primary sclerosing cholangitis.

**Table 1 jcm-09-00319-t001:** Baseline characteristics of patients listed for transplant with autoimmune liver disease (*n* = 26,432).

	AIH (*n* = 7412)	PBC (*n* = 8119)	PSC (*n* = 10,901)	*p* Value
Age (mean ± SD)	44.6 ± 16.3	54.9 ± 10	45.6 ± 14.3	<0.001
Sex (%male)	24.5	14	66.9	<0.001
Diabetic (%)	11.5	7.5	5.9	<0.001
Blood Type (%)				<0.001
A	35.6	37.2	38.7	
B	12.4	11.1	12.5	
AB	3.5	3.3	3.5	
O	48.4	48.3	45.2	
MELD (mean ± SD)	18.9 ± 9.5	17.1 ± 7.9	16.4 ± 7.9	<0.001
Albumin in g/Dl (mean ± SD)	3 ± 0.7	3 ± 0.6	3.1 ± 0.7	<0.001
Removed from waitlist for being too sick or death (%)	21.8	24.4	15.6	<0.001
Transplanted (%)	52.3	59.1	65.9	<0.001

AIH: autoimmune hepatitis; PBC: primary biliary cholangitis; PSC: primary sclerosing cholangitis.

**Table 2 jcm-09-00319-t002:** Cause of death while on liver transplant waitlist by etiology. AIH: autoimmune hepatitis; PBC: primary biliary cholangitis; PSC: primary sclerosing cholangitis.

	AIH (*n* = 1115)	PBC (*n* = 1424)	PSC (*n* = 1108)	*p* Value
Died on waitlist (%)	15	17.5	10.2	<0.001
Cause of death (%)				0.02
Liver-related	6.6	5.7	6.9	
Cardiac	7.5	8.3	5.9	
Pulmonary	3.4	3.3	1.3	
Renal	0.8	0.7	0.3	
Infectious	17.2	15.3	17.2	
Malignancy	1.3	0.6	2.6	
Other	63.2	66	65.9	

**Table 3 jcm-09-00319-t003:** Cox regression analysis for time to death or waitlist removal for clinical deterioration. REF: reference; HR: hazard ratio; NS: not significant; MELD: model for end stage liver disease; AIH: autoimmune hepatitis; PBC: primary biliary cholangitis; PSC: primary sclerosing cholangitis.

	HR	95%CI	*p* Value	HR	95%CI	*p* Value
Age	1.04	1.04 to 1.04	<0.001	1.04	1.04 to 1.05	<0.001
Sex(male)	0.75	0.71 to 0.8	<0.001	0.73	0.67 to 0.8	<0.001
Presence of diabetes	1.73	1.57 to 1.91	<0.001	1.18	1.07 to1.31	<0.001
Initial MELD	1.13	1.12	<0.001	1.13	1.13 to 1.14	<0.001
Initial albumin(g/dL)	0.47	0.44 to 0.49	<0.001	0.68	0.64 to 0.72	<0.001
UNOS Region						
1	**REF**			**REF**		
2	0.95	0.74 to 0.97	0.02	0.98	0.8 to 1.19	NS
3	0.97	0.83 to1.13	NS	0.9	0.73 to1.12	NS
4	0.97	0.94 to 1.11	NS	1	0.83 to1.13	NS
5	0.95	0.83 to 1.08	NS	0.94	0.78 to1.13	NS
6	1.07	0.89 to 1.28	NS	0.97	0.76 to 1.23	NS
7	0.75	0.65 to 0.86	<0.001	0.73	0.6 to 0.91	0.04
8	0.95	0.83 to 1.1	NS	0.92	0.76 to 1.13	NS
9	0.92	0.71 to0.95	0.01	0.89	0.72 to 1.11	NS
10	1.15	0.99 to 1.33	NS	1.14	0.93 to 1.4	NS
11	0.98	0.85 to 1.13	NS	1.12	0.91 to 1.38	NS
Blood type						
A	**REF**			**REF**		
B	0.9	0.82 to 0.99	0.03	0.85	0.75 to 0.97	0.02
AB	0.87	0.73 to 1.05	NS	0.91	0.7 to 1.18	NS
O	1.03	0.97 to 1.09	NS	0.95	0.88 to 1.03	NS
Etiology						
AIH	REF			REF		
PBC	1.2	1.11 to 1.27	<0.001	1.11	1.01 to 1.21	0.03
PSC	0.77	0.71 to 0.82	<0.001	0.93	0.84 to 1.03	NS

**Table 4 jcm-09-00319-t004:** Competing risk analysis of time to death or waitlist removal for clinical deterioration with transplant as competing risk.

	SHR	95%CI	*p* Value	SHR	95%CI	*p* Value
Age	1.03	1.03 to 1.04	<0.001	1.03	1.03 to 1.04	<0.001
Sex(male)	0.69	0.65to 0.73	<0.001	0.76	0.69 to 0.83	<0.001
Presence of diabetes	1.4	1.27 to 1.54	<0.001	1.13	1.02 to 1.25	0.02
Initial MELD	1.03	1.02 to 1.03	<0.001	1.02	1.02 to 1.03	<0.001
Initial albumin(g/dL)	0.69	0.66 to0.73	<0.001	0.75	0.71 to 0.8	<0.001
UNOS Region						
1	REF					
2	0.79	0.69 to 0.9	0.001	0.78	0.64 to 0.95	0.01
3	0.45	0.39 to 0.53	<0.001S	0.37	0.3 to 0.46	<0.001
4	0.84	0.73 to 0.96	0.01	0.86	0.72 to 1.04	NS
5	0.92	0.81 to 1.04	NS	0.89	0.75 to 1.07	NS
6	0.87	0.73 to 1.04	NS	0.84	0.66 to 1.08	NS
7	0.63	0.55 to 0.73	<0.001	0.61	0.49 to0.75	<0.001
8	0.74	0.64 to0.86	<0.001	0.76	0.62 to 0.93	0.01
9	0.8	0.69 to 0.93	0.003	0.77	0.62 to 0.95	0.02
10	0.75	0.65 to 0.86	<0.001	0.63	0.51 to 0.77	<0.001
11	0.71	0.61 to 0.82	<0.001	0.64	0.52 to 0.78	<0.001
Blood type						
A	REF					
B	0.84	0.77 to 0.93	<0.001	0.71	0.62 to 0.81	<0.001
AB	0.65	0.54 to 0.79	<0.001	0.51	0.39 to 0.66	<0.001
O	1.1	1.04 to1.16	0.02	0.93	0.86 to 1.01	NS
Etiology						
AIH	REF					
PBC	1.09	1.02 to 1.17	0.001	0.94	0.85 to 1.03	NS
PSC	0.67	0.63 to 0.72	<0.001	0.8	0.72 to 0.89	<0.001

REF: reference; SHR: subhazard ratio; NS: not significant; MELD: model for end stage liver disease; AIH: autoimmune hepatitis; PBC: primary biliary cholangitis; PSC: primary sclerosing cholangitis.

**Table 5 jcm-09-00319-t005:** Trend in proportion of patients listed over time by autoimmune liver disease.

	1987–1995	1996–2001	2008–2016
AIH (%)	26.5	27	30
PBC (%)	37.6	31.7	27.3
PSC (%)	35.9	41.3	42.7
Total (*n*)	24841	13,896	9695
			*p* = 0.175

AIH: autoimmune hepatitis; PBC: primary biliary cholangitis; PSC: primary sclerosing cholangitis.

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
