# Peer review of "Mortality on the UNOS Waitlist for Patients with Autoimmune Liver Disease"

_jcm, 2020, doi:10.3390/jcm9020319_

Round 1

Reviewer 1 Report

The study is of interest. The statistical methods are sophisticated. The results are coherent. The study is very well written. No further comments. 

Author Response

No comments to address. Thank you.

Reviewer 2 Report

The present study aims to investigate differences in waitlist mortality of autoimmune liver diseases (AIH, PBC, and PSC) using the UNOS database. The authors identified patients who were listed for liver transplantation from 1987 to 2016 with a primary diagnosis of AIH, PBC, or PSC. They conclude that patients with PBC and AIH were more likely than PSC to be removed from the waitlist for death or clinical deterioration.

The issue is of interest and the paper is methodologically sound; I only have a few comments:

- The authors have recently published a very similar paper elsewhere: Increased Posttransplant Mortality for Autoimmune Hepatitis Compared With Other Autoimmune Liver Diseases. J Clin Gastroenterol. 2019 Oct 23. (doi: 10.1097/MCG.0000000000001271.) Please comment on the discussion and highlight the differences.

- It would be interesting to see if there was a different trend in more recent years compare to the ’80; I suggest the authors include a comparison between the two decades included in the study (1987-1996 and 1997-2016).

- References: I suggest the authors to include a more complete citation of the literature

Minor:

- Discussion: “Female predominance in the AIH and PBC groups suggests that gender may influence adverse waitlist outcomes as well” (Page 6, line 143). Since AIH and PBC are well known female predominant diseases, I don't believe the authors can draw that conclusion. Please remove the sentence.

- Please use abbreviations throughout the paper. I.e. Page 6, line3 164 use PSC instead of “Primary sclerosing cholangitis”

Author Response

Dear Reviewer,

Thank you for your timely and thoughtful comments. We have attempted to address each one to the best of our abilities. Please see below.

The authors have recently published a very similar paper elsewhere: Increased Posttransplant Mortality for Autoimmune Hepatitis Compared With Other Autoimmune Liver Diseases. J Clin Gastroenterol. 2019 Oct 23. (doi: 10.1097/MCG.0000000000001271.) Please comment on the discussion and highlight the differences.

Response: Thank you for this comment. Although we appreciate you highlighting other work done by this group, a comparison here would be out of the intended scope of this specific manuscript we are submitting. Our focus here was solely on the pre-transplant morbidity and mortality and the associated factors for patients that are listed with autoimmune liver disease; specifically looking at AIH. Post-transplant outcomes carry with them a somewhat different set of complications and possibilities that are not all pertinent to the key points of our current submission.

- It would be interesting to see if there was a different trend in more recent years compare to the ’80; I suggest the authors include a comparison between the two decades included in the study (1987-1996 and 1997-2016).

Response: Thank you for your recommendation. We have added a new table, Table 5 (see attachment), that specially addresses this. We have also added some comments in regards to this in the discussion section. See page 7, line 178.

- References: I suggest the authors to include a more complete citation of the literature 

Response: We have reviewed the manuscript for any un-cited claims and ensured its completeness. This area of research is still relatively devoid of a robust volume of literature. We have tried to include all relevant material.

Minor:

- Discussion: “Female predominance in the AIH and PBC groups suggests that gender may influence adverse waitlist outcomes as well” (Page 6, line 143). Since AIH and PBC are well known female predominant diseases, I don't believe the authors can draw that conclusion. Please remove the sentence.

Response: Thank you for this comment. This claim has been removed.

- Please use abbreviations throughout the paper. I.e. Page 6, line3 164 use PSC instead of “Primary sclerosing cholangitis”

Response: Thank you for noting this. We have gone back to include the abbreviations.

Reviewer 3 Report

There only some minor comments:

In table 3, regions are shown with numbers (1, 2, 3 to 11), without an explanation what are these numbers represent. In page 6 (1st paragraph) the authors present that they examined temporal trends in liver transplant listing by autoimmune etiology. It will be more illustrative to present these trends also schematically. They should also discuss why these trends are noticed.  One of the probable explanation for the difference noticed in mortality between PBC and PSC is the allotment of MELD exception points. The authors should describe which are these exception points.

Author Response

In table 3, regions are shown with numbers (1, 2, 3 to 11), without an explanation what are these numbers represent.

Response: Thank you for noting this. The “regions” being referenced are the UNOS regions for organ procurement and transplantation. We have changed the title to say, “UNOS Region” for both table 3 and table 4. Please see uploaded attachments. Tables have been inserted in the text for the editors to format per their discretion.

One of the probable explanation for the difference noticed in mortality between PBC and PSC is the allotment of MELD exception points. The authors should describe which are these exception points. 

Response: Thank you for the comment. This has been highlighted in the discussion section, page 7. We discuss the disbursement of exception points for occurrence of cholangitis in patients with PSC and how it affects their listing mortality in relation to PBC.

In page 6 (1st paragraph) the authors present that they examined temporal trends in liver transplant listing by autoimmune etiology. It will be more illustrative to present these trends also schematically. They should also discuss why these trends are noticed.  

Response: Thank you for your recommendation. We have added a new table, Table 5, that specially addresses this. We have also added some comments in regards to this in the discussion section. See page 7, line 178.

Round 2

Reviewer 2 Report

I believe the paper can be published in the present form